# Leadership Styles in Non-Profit Institutions: An Empirical Study for the Validation and Reliability of a Scale in the Latin American Context

**DOI:** 10.3390/bs15020130

**Published:** 2025-01-26

**Authors:** Javier Enrique Espejo-Pereda, Elizabeth Emperatriz García-Salirrosas, Miluska Villar-Guevara, Israel Fernández-Mallma

**Affiliations:** 1UPG de Ciencias Empresariales, Escuela de Posgrado, Universidad Peruana Unión, Lima 15102, Peru; javier.espejo@upeu.edu.pe; 2Faculty of Management Science, Universidad Autonoma del Peru, Lima 15842, Peru; egarciasa@autonoma.edu.pe; 3EP de Administración, Facultad de Ciencias Empresariales, Universidad Peruana Unión, Juliaca 21100, Peru; 4EP de Educación, Facultad de Ciencias Humanas y Educación, Universidad Peruana Unión, Juliaca 21100, Peru; pastorisrael@upeu.edu.pe

**Keywords:** leadership styles, promoting leadership, validity, reliability, scale, non-profit

## Abstract

There is no doubt that leadership is one of the most researched and disseminated topics in recent years, and over time, some distinguished models have developed a solid foundation and a reputable structure. From this perspective, this study analyzes the evidence of validity and reliability of a scale that assesses leadership styles in non-profit institutions. The study had an instrumental design. The sample consisted of 272 workers from nine Latin American countries, aged between 19 and 68 years (M = 34.08 and SD = 8.61), recruited through non-probabilistic sampling. A validity and reliability analysis of the scale confirmed the nine items and three original factors (servant, empowering and shared leadership). The KMO test reached a high level (0.898 > 0.70), and the Bartlett test reached a highly significant level (Sig. = 0.000). The scale also showed good internal consistency (α = 0.918 to 0.956; CR = 0.918 to 0.957; AVE = 0.755 to 0.880). Likewise, for the Confirmatory Factor Analysis, a measurement adjustment was performed, obtaining excellent and acceptable fit indices for Model 2 (CMIN/DF = 1.794; CFI = 0.993; SRMR = 0.023; RMSEA = 0.054; Pclose = 0.369). This study provides a brief and useful tool to measure leadership styles in Latin America, as a scale used specifically for this context would allow for a more accurate and valid assessment. This is crucial for generating effective organizational interventions, fostering the development of authentic leaders, and improving the competitiveness of non-profit institutions.

## 1. Introduction

Leadership has become one of the most interesting and prevalent topics in the academic and business world ([45]). This skill is used to build trust between the leader and team members ([14]). As history itself tells us, leadership has existed since ancient times, that is, since the time of the Sumerians, Egyptians, Babylonians, among other great civilizations ([28]). All of them, guided by a leader, presided over large crowds, directed the government of a nation, controlled wars and promoted collective ideas, being highly educated in the practice of this skill ([28]; [74]). Since modern society is undergoing significant economic, social, cultural, political and technological changes, it is in this context that the study of leadership takes on special importance, since these phenomena are related to the orientation of people and their influence on the achievement of an organization’s goals ([54]).

A leader is seen as the heart of an organization and cannot be underestimated ([4]). Leadership is recognized by spreading ideas, thoughts or actions. However, according to [26] ([26]), a leader is one who solves problems through creative ways to achieve sustainable success. Researchers have demonstrated and proposed several forms of leadership ([5]), and with its evolution, renowned authors such as James Burns, Bernard Bass, Steven Covey, among others, have stood out and contributed to the study of leadership styles ([64]); each style are methods or patterns of behavior that the leader adopts to manage his organization effectively ([78]). It is important to note that different leadership styles seem to work for most leaders in different situations, leading to the claim that there is no best leadership style and that the most successful leaders tend to adopt most or all of the different styles that exist so far ([48]; [59]; [78]).

In this sense, business leaders are seen as agents of change in the world of work ([32]; [42]; [46]; [47]; [69]). These leaders must adopt an effective leadership style, serving, sharing and empowering their employees ([21]). Reliability, competence, communication, coordination and employee commitment reflect the success of the leadership style in this new era ([25]). At the same time, those responsible for human talent management must adopt appropriate methods to ensure high performance of the work group. These three leadership styles relate to talent management from various perspectives, such as employee performance ([14]), recognition ([76]), motivation ([54]), love for the brand ([48]), effectiveness ([73]), ecological behavior of employees ([81]), organizational innovation ([57]), spirituality ([65]), agile team productivity ([41]), passion at work ([68]), and psychological safety ([36]), among others.

After carefully reviewing the background, there is evidence of a growing interest in continuing to research this topic among academics and professionals in the business sector. Previous studies have been applied to various areas, sectors and populations, such as business, management and accounting, social sciences, psychology, economics, econometrics and finance. Bibliometric indicators show the ten countries with the greatest dissemination of their scientific results: The United States, Malaysia, Indonesia, United Kingdom, India, Australia, China, Iran, Spain and Turkey. When evaluating scientific dissemination by country, it has been found that it is necessary to expand the research carried out in Latin American contexts, given that more scientific literature can support and guide future research in this line of knowledge.

In light of the above, this study addresses an important research gap. Given that non-profit institutions provide a nontraditional business environment, assessing user perceptions of servant, empowered and shared leadership translate as urgent. Few studies have focused on the leadership of influential non-profit institutions in Latin America, which makes this study valuable. In addition, the absence of previous studies that have employed an inclusive sample of nine Latin American countries generates the need for a psychometric scale that assesses leadership styles adapted to this region of the world, given that this context is characterized by its cultural diversity, unequal socioeconomic structures and specific organizational dynamics. Traditional leadership theories and models, often developed in Western contexts, do not always capture the cultural and contextual particularities of Latin America, such as the impact of collectivization, the power of hierarchies and the influences of interpersonal trust. A scale used specifically for this context would allow for a more accurate and valid assessment. This is crucial for generating effective organizational interventions, fostering the development of authentic leaders and enhancing the competitiveness of non-profit institutions in the region. Considering the above, the objective of the present study was to analyze the evidence of validity and reliability of a scale that evaluates leadership styles in non-profit institutions.

## 2. Literature Review

### 2.1. Leadership Styles

Research in this field is diverse and has addressed the different leadership styles, based on the various theories proposed since the beginning of the 20th century ([6]). Empirical studies cover leadership models such as: ethical ([17]), authentic ([8]), server ([77]), transactional ([30]), situational ([6]), transformational ([72]), democratic ([62]), autocratic ([5]), laissez-faire ([58]), strategic ([71]), bureaucratic ([15]), charismatic ([37]), people-oriented ([49]), task-oriented ([18]), and relationship-oriented ([44]), among others. Leadership styles can influence the creativity of leaders in front of their team, and this in turn influences other members of the work group ([4]). From this variety, the present study reviews servant, empowering and shared leadership.

#### 2.1.1. Servant Leadership

Servant leadership is known as a comprehensive and altruistic leadership style ([48]) in which leaders focus primarily on the wants and needs of their team ([63]); instead of serving his superiors, the servant leader emphasizes the benefits for his work team and the community at large ([77]). This is a field of knowledge that deserves careful study due to its numerous positive results in work groups ([29]) and its commitment to high human-centered moral and ethical standards ([33]). In 1970, Robert Greenleaf argued that the most accurate test of servant leadership is a conscious decision to be a servant first and a natural response to the desire to lead ([34]). A servant leader is influential; he knows how to win the favor of his team before asking them to follow his ideals because, deep down, everyone wants to be valued; however, the influence of a leader is created by making ethical decisions and building good relationships with others ([17]).

#### 2.1.2. Empowering Leadership

Empowering leadership is often defined in two ways. First, empowering leadership includes the behaviors of a formal leader or a leader who holds a position and power in an organization ([61]), such as encouraging your team to express opinions and ideas, facilitating collaborative decision-making, supporting information sharing and teamwork ([70]). Secondly, recent research conceptualizes empowering leadership as a method of sharing power among leaders that automatically increases people’s autonomy (both individually and as a group) and distinguishes a more proactive participation in their workplace ([75]). Although there are many definitions of empowering leadership, there are differences in the dimensions according to their theoretical model ([40]). However, management specialists have also studied the construct in relation to other factors, reporting that it is sufficiently distinct from these leadership constructs to warrant a unique line of academic research ([70]).

#### 2.1.3. Shared Leadership

Shared leadership is a management model in which responsibility and decision-making authority are distributed among all members of a team or group, rather than concentrated in a single person ([10]). This approach fosters cooperation, empowerment and active participation of all actors and uses different skills and knowledge to achieve common goals more effectively ([23]). Shared leadership is an organizational dynamic in which leadership is not limited to one authoritarian person but is distributed among several individuals who assume leadership roles based on their specific capabilities and circumstances ([79]). This type of leadership fosters collective responsibility, innovation and commitment, thereby increasing organizational adaptability and resilience ([82]).

### 2.2. Scales to Evaluate Leadership Styles

In discerning the theoretical proposals based on leadership styles, important scientific contributions have been found that argue that this construct is linked to organizational commitment, sustainability, sustainable performance, job satisfaction, burnout, innovative behavior, well-being, empowerment, trust and team cohesion. Therefore, an analysis of the studies already reported on this topic reveals its importance and the need to provide a valid metric that can help assess leadership styles in non-profit institutions and that can be used as a strategy to improve organizational performance and identify behavioral patterns. A well-designed tool can provide objective data that reveals strengths and areas for improvement in leadership, facilitating the implementation of more inclusive and people-oriented strategies. This not only promotes a positive work environment that increases staff satisfaction and retention, but also aligns leadership efforts with the institutional mission, ensuring transparency and accountability. It also fosters a strong and coherent organizational culture, which translates into higher quality services offered to beneficiaries, consolidating the trust and sustainability of the institution in the long term. To this end, it is expected that all measurement instruments meet valid psychometric properties so they can be used in different contexts. In this regard, a summary of the measurement scales published in high-impact journals is detailed in Table 1, where, in addition, the factors, population and other characteristics to be considered in the analysis have been clearly described.

After this review of the scales already published on the topic under study, a growing interest in having a metric that helps evaluate leadership styles in workers of non-profit entities for the context of Latin America has been validated. Previous studies have provided criteria for this construct in countries such as the United States, Brazil, Spain, South Africa, Angola and Indonesia. Previous scales measuring this construct have been applied to diverse groups and populations, including the following examples: employees of small and medium-sized enterprises (SMEs), workers of public entities, workers of private entities, team leaders of cooperatives, workers of non-profit organizations, directors and executives of private institutions, principals and teachers of secondary schools and members of the Public Relations Society of America (PRSA). 

## 3. Materials and Methods

### 3.1. Study Design and Participants

The design of this study was instrumental ([9]). The population group consisted of workers from a non-profit agency, belonging to nine Latin American countries. This institution provides humanitarian aid and development in 130 countries around the world and has a very important impact in Latin America. The inclusion criteria were to be of legal age (minimum 18 years) and to have more than one year working in management, leadership, administrative and technical roles in project implementation. To select the participants, a non-probabilistic convenience sampling was applied, choosing those who were available and willing to participate in the research. This was obtained through a request for participation through the directors of each agency–country. A total of 272 workers between the ages of 19 and 68 participated (M = 34.08 and SD = 8.61). The majority of the participants were women (62.9%), with an age range of 19 to 30 years (48.2%), who worked at the Ecuador Country Agency (54%) and who had been working for the institution between 1 to 5 years (81.6%) (Table 2).

### 3.2. Ethical Considerations

The research was previously evaluated and approved by the Ethics Committee of a private university in Peru (2023-CE-EPG-00153). Subsequently, during the period between January and May 2024, participants were invited to complete an online questionnaire through Google Forms. Prior to the data collection process, this study followed the principles of the Declaration of Helsinki ([52]; [66]) and the rules of confidentiality, notifying each participant about the purpose of this research and collecting their informed consent, which was declared under the following affirmation: *“I acknowledge that by filling out this questionnaire I am giving my consent to participate in the study”*.

### 3.3. Measurement Scale and Back-Translation

The questionnaire technique was used as a data collection instrument ([51]). The online questionnaire was structured in three sections: the first section included the instructions and informed consent of the participants; the second section included the sociodemographic variables; and the last contained the leadership styles instrument. To measure the construct, a short instrument of nine items and three factors was used (Table 3) based on the study by [21] ([21]), which initially consisted of six items. However, in the back-translation process, ambiguities were found in the items for shared leadership (SD), which became even more evident when the pilot test was performed, finding that the items had low factor loadings and low internal consistency. Thus, it was decided to replace these two items with four items from the [22] ([22]) scale, leaving the instrument as follows: two items for servant leadership (SL) ([21]; [67]), three items for empowering leadership (EL) ([21]; [61]), and four items for shared leadership (SD) adapted from recent research ([22]). A five-point Likert-type response format was used to respond to each item, measured from 1 (“Strongly disagree”) to 5 (“Strongly agree”). Internal consistency was measured by Cronbach’s alpha (α = 0.918 to 0.956).

The back-translation method with bilingual tests was used to translate the original English version into Spanish. Two bilingual speakers (Spanish–English) whose first language was Spanish individually completed the translation from English to Spanish. A focus group composed of six collaborators representing the participating countries, meeting the inclusion criteria of the study, compared, discussed and modified the translations to obtain the improved version of the instrument in Spanish, ensuring that the items were understandable in the context of each country (Peru and Brazil). The instrument was also translated from Spanish to Brazilian Portuguese by a native speaker and reviewed by a professional in the area to ensure the understanding of the instrument before its administration.

### 3.4. Data Analysis

In order to meet the overall objective of this study, two statistical softwares were used for data analysis. The first was to evaluate Exploratory Factor Analysis (EFA) using SPSS. Secondly, Covariance Structural Equation Modeling (CB-SEM) was used to perform Confirmatory Factor Analysis (CFA) to assess convergent and discriminant reliability and measurement model fit using AMOS software version 24.

## 4. Results

The descriptive statistical results of the items, such as the mean, standard deviation, skewness and kurtosis of the scale are detailed in Table 4. In addition, it is noted that the values of skewness and kurtosis values are mostly less than ±1.5 ([31]), except for items SD1 and SD2 of the kurtosis column, which showed a slight non-compliance with the multivariate normality of the data, for which the maximum likelihood method was used.

### Exploratory Factor Analysis

The Exploratory Factor Analysis (EFA) of the items is detailed in Table 5, which shows that the items are divided into three factors according to the variable under study. In addition, there is a clear difference between the three factors. The KMO and Bartlett test (Kaiser-Meyer-Olkin sample adequacy measure = 0.898 greater than 0.70) is high, and the Bartlett test (Sig. = 0.000) is very significant to perform the factor analysis. The total variance explained in the model is 83.369%, which is greater than 50%, being shared leadership (SD) = 68.163%, empowerment leadership (EL) = 9.269% and service leadership (SE) = 5.937%. All items were grouped according to their initial dimensions. Then, the Confirmatory Factor Analysis (CFA) was carried out.

Next, the validation of the final measurement model with convergent reliability and validity is detailed. The Cronbach’s Alpha (α) values are between 0.918 and 0.956, as can be seen in Table 6. These values are satisfactory since, for the model to be considered at an adequate level, all values must be above 0.70 ([1]). Likewise, the composite reliability (CR) values are between 0.918 and 0.956, which is also favorable since, for it to be considered an optimal model, the values must be greater than 0.60 ([11]). On the other hand, the AVE values are between 0.755 and 0.880, which is considered optimal since, to have acceptable values for this indicator, they must be equal to or greater than 0.50 ([35]). This means that the measurement model meets all indicators of reliability and convergent validity.

Figure 1 shows the factor structure of the scale to assess leadership style in non-profit institutions.

The fit indicators of the measurement model of the scale to assess leadership style are detailed in Table 7 according to the results of the CFA with a three-dimensional structure where the nine items explained the three factors (Model 1); however, not all the goodness of fit was excellent, so the model was re-specified based on the modification index (MI) ([19]). In this sense, due to the similarity in phrasing, there was a correlation between errors e6 and e7, which correspond to items SD4 and SD3, respectively, so a model adjustment was made, giving rise to Model 2, obtaining excellent fit indices (Table 7).

To evaluate the discriminant validity of the model, the Fornell–Larker criterion was used, thus, the square root of the AVE of each factor was calculated, which had to be greater than the highest correlation between the factors of the measurement model ([35]). Table 8 shows that all values in the bold diagonal are greater than the correlations. Additionally, the Heterotrait–Monotrait (HTMT) criterion has been taken into account in this study ([38]). If the HTMT value is less than 0.90, it is considered that there is discriminant validity between two reflective constructs. In this sense, Table 8 shows that the highest correlation has a value of 0.755, which is less than 0.90. With these results, the discriminant validity of the model is fulfilled.

## 5. Discussion

Leaders adopt various leadership styles by training or empirically, which has an impact on the organizations they lead depending on the style adopted. Although studies have addressed the advantages and consequences of leadership styles, which are related to the personality and characteristics of the individual, our study contributes to the search for identifying three leadership styles: servant leadership, empowering leadership and shared leadership. Thus, the objective of the present study was to analyze the evidence of validity and reliability of a scale that assesses leadership styles in non-profit institutions.

This research has an impact, not only because it adapts a scale to analyze three leadership styles from a Latin American population, but also in non-profit contexts. This type of study constitutes an advance in the process of broadening the spectrum of research, which has generally focused on North America and Europe ([7]; [39]; [43]), and opens the possibility for comparative studies of the same leadership styles in underexplored latitudes such as Latin America. Seen from the perspective of the need to internationalize the studies, and given that bibliometrics supports the need to replicate in other latitudes studies conducted in English-speaking contexts, the relevance of the present research is considered ([2]; [12]; [20]; [43]).

The scale was taken from the study by [21] ([21]), which analyzes the three factors in six items: two for servant leadership (SL), two for empowering leadership (EL), and two for shared leadership (SD). However, since the shared leadership (SD) items did not contribute to the structural model, it was decided to remove them and include four items from the [22] ([22]) study. According to [24] ([24]) the items should have a common core that increases reliability, and that is semantically clear and specific. The authors’ decision to include four items from another scale is justified by the ambiguity of the items of the original scale at the time of back-translation. Likewise, according to [16] ([16]), there is no reason why one item should be conceptualized as superior to the others, but on the contrary, they are essentially replaceable, especially if the authors consider that a given item better captures or more faithfully represents the construct under study ([27]).

Although scales have been found that measure different leadership styles ([53]; [54]; [60]), the authors chose to adapt the aforementioned versions because they were closer to the study population, workers in non-profit institutions. On the other hand, although the scale is made up of nine items, this does not affect the level of reliability, since item theory (IRT) holds that it is possible to measure constructs with few items without affecting the reliability of the instrument ([55], [56]), as well as the fact that other authors sustain the validity of scales even with a single item ([80]). It is evident that there is a clear debate regarding the best methodology for item selection, and all of these should conform to conventional measurement theory ([27]); however, the authors consider that this adaptation is within the parameters established for the adaptation and validation of a scale.

The Exploratory Factor Analysis yielded a three-factor solution, where the Cronbach’s Alpha values of each construct ranged between 0.918 and 0.956, evidencing the high reliability of the measurement scale, higher than the values reported by the original measure. The composite reliability (CR) also shows favorable scores (between 0.918 and 0.956), and the AVE values are optimal (between 0.755 and 0.880). Regarding the goodness of fit, a model adjustment was needed because there were correlations between the errors of items SD4 and SD3 (Shared Leadership items); thus, a model adjustment was made giving rise to Model 2, where excellent fit indices were obtained (CMIN/DF = 1.794; CFI = 0.993; SRMR = 0.023; RMSEA = 0.054; Pclose = 0.369). Additionally, the discriminant validity also obtained acceptable scores.

Regarding the study carried out by [53] ([53]), the reliability and discriminant validity of the instrument used are not evident. On the other hand, the study by [60] ([60]) analyzed leadership from two perspectives, dividing the 15 items between leadership focused on people and leadership focused on production. Thus, the present instrument did not find any similar instrument for comparative analysis; however, the statistical results show its solidity and reliability for use in organizational contexts of the third sector in Latin America. Being an instrument with 9 items, it is positioned as a very practical tool in its application, and its adaptation to the Latin American context allows for a deeper study of leadership styles in organizations and guides managerial decision-making.

### 5.1. Theoretical and Managerial Implications

This research also reports theoretical and practical implications. First, it contributes to the literature on the line of knowledge of leadership in non-profit organizations by providing a scale that measures the influence of leadership style on organizational performance. Currently, leadership styles and their contribution to organizational efficiency and effectiveness are theoretically disseminated; however, this study has allowed the development of a reliable tool to implement organizational strategies for the application of leadership styles that contribute to a positive impact on the fulfillment of the mission of non-profit institutions and the lives of the people served by the organization itself.

The study will allow third sector organizations to have greater knowledge and confidence in leadership styles in order to apply effective leadership that results in an increase in the satisfaction of donors, beneficiaries and staff, positively impacting organizational innovation and sustainability. Senior management can use this scale to measure the effectiveness of leadership styles for the third sector, considering that leadership styles define organizational culture, and effective leadership can generate important changes within third sector organizations since they base their actions on humanitarian principles; therefore, a leader with a positive leadership style has an impact on increasing organizational well-being, promoting a culture of quality and service.

### 5.2. Limitations and Future Research

Despite having a valuable contribution to the academic community and to non-profit organizations, this research reports some limitations. First, the focus was limited to the study of leadership styles in the Latin American context. Therefore, future studies may include non-governmental organizations from other continents and cultures in order to compare leadership styles in different regions of the world. Future research could study demographic differences in which the effects of age and gender on the adoption of a leadership style could also be analyzed. In order to identify the diversity of leadership and organizational cultures, for future studies it is suggested to include a variety of non-profit institutions and not only an international organization based in different countries; likewise, the need has been identified to compare the perception that workers have about the leadership style of their supervisors or immediate bosses from different business sectors.

## 6. Conclusions

The main objective of the study was to analyze the validity and reliability evidence of a scale that assesses leadership styles in non-profit institutions. The scientific community is provided with a fully reliable, flexible and practical scale to be applied by professionals and academics in this field of knowledge. Therefore, having a scale that measures leadership styles is essential since it provides a valid tool that can be used as a strategy to improve organizational performance, develop a more people-oriented leadership style, increase staff satisfaction and retention, ensure compliance with the institutional mission, guarantee transparency and accountability, strengthen organizational culture and improve the quality of service to beneficiaries. Having a valid measurement scale will provide a solid basis for making informed and strategic decisions that can lead to a more significant and sustainable social impact.

In this sense, the validity and reliability analysis of the scale confirmed the nine items and three original factors (servant, empowering and shared leadership). The KMO test reached a high level (0.898 > 0.70), and the Bartlett test reached a highly significant level (Sig. = 0.000). The scale also showed good internal consistency (α = 0.918 to 0.956; CR = 0.918 to 0.957; AVE = 0.755 to 0.880). Likewise, for the Confirmatory Factor Analysis, a measurement adjustment was performed, obtaining excellent and acceptable fit indices for Model 2 (CMIN/DF = 1.794; CFI = 0.993; SRMR = 0.023; RMSEA = 0.054; Pclose = 0.369). In this sense, this study provides a short and useful tool to assess leadership styles in a user-friendly way to be applied in non-profit institutions in Latin America. This study is considered an important contribution to senior management and related areas of the business environment in the context of non-profit entities.

## Figures and Tables

**Figure 1 behavsci-15-00130-f001:**
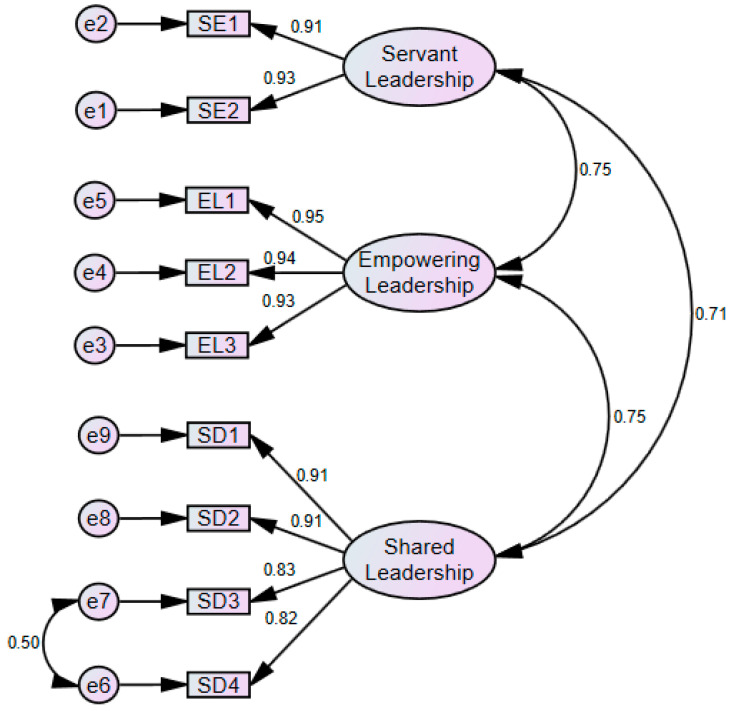
Factor structure of the scale to assess leadership style.

**Table 1 behavsci-15-00130-t001:** Measurement scales for leadership styles.

Scale	Author(s)	Country	Items	Factors	Population	α
Leadership Styles Hetero-assessment Scale	[60] ([60])	Brazil	11	(1) Focus on people; (2) Focus on results.	Professionals	0.83 to 0.90
Multifactor Leadership Questionnaire (MLQ)	[13] ([13])	USA	72	(1) Transformational leadership; (2) Transactional leadership; (3) Laissez-faire leadership.	Private, public and non-profit organizations.	0.69, 0.87 and 0.91
Multifactor Leadership Questionnaire (MLQ)	[78] ([78])	South Africa	35	(1) Transformational leadership; (2) Transactional leadership; (3) Laissez-faire leadership.	Secondary school principals and teachers	NE
Multifactor Leadership Questionnaire (MLQ)	[54] ([54])	Angola	10	(1) Transformational leadership; (2) Transactional leadership; (3) Laissez-faire leadership.	Directors and executives of private institutions	0.803
Multifactor Leadership Questionnaire (MLQ)	[53] ([53])	Spain	28	(1) Transformational leadership; (2) Transactional leadership.	Cooperative team leaders	Between 0.802 and 0.915
Multifactor Leadership Questionnaire (MLQ)	[50] ([50])	Spain	45	(1) Transformational leadership; (2) Transactional leadership; (3) Laissez-faire leadership.	Workers from various sectors	0.90, 0.80 and 0.70
Communication and Public Relations Leadership Survey	[3] ([3])	USA	14	(1) Transactional leadership; (2) Transformational leadership; (3) Pluralistic leadership.	Members of the Public Relations Society of America (PRSA)	0.668
Leadership Styles Scale	[21] ([21])	Indonesia	6	(1) Servant leadership; (2) Empowering leadership; (3) Shared leadership	Employees of small- and medium-sized companies	0.93

Note: NE = Not Specified.

**Table 2 behavsci-15-00130-t002:** Sociodemographic characteristics of the participants (n = 272).

Category		Frequency	%
Sex	Female	171	62.9
	Male	101	37.1
Age range	19–30 years	131	48.2
	31–40 years	82	30.1
	41–50 years	45	16.5
	51–68 years	14	5.2
Agency–Country where you work	Ecuador	147	54.0
	Peru	69	25.4
	Chile	26	9.6
	Brazil	19	6.9
	Venezuela	4	1.5
	Mexico	3	1.1
	Colombia	2	0.7
	Argentina	1	0.4
	Paraguay	1	0.4
Years of service	1–5 years	222	81.6
	6–10 years	31	11.4
	11–15 years	5	1.8
	16 years or more	14	5.2

**Table 3 behavsci-15-00130-t003:** Measurement scale in Spanish with an English translation.

Predictor	Code	Item
Servant Leadership (SE)		*En la organización donde laboro, mi jefe inmediato me apoya para… [In the organization where I work, my immediate boss supports me to…]*
	SE1	Aumentar mis competencias en el trabajo. [Increase my skills at work].
	SE4	Mejorar mis habilidades de comunicación. [Improve my communication skills].
Empowering Leadership (EL)		*En la organización donde laboro, mi participación en la toma de decisiones organizacionales… [In the organization where I work, my participation in organizational decision-making…]*
	EL1	Es considerada. [It is considered].
	EL2	Es valorada. [It is valued].
	EL3	Es promovida. [It is promoted].
Shared Leadership (SD)		*En la organización donde laboro, los miembros de mi equipo directivo… [In the organization where I work, the members of my management team…]*
	SD1	Planifican colectivamente las operaciones. [They collectively plan operations].
	SD2	Se alientan mutuamente en el trabajo. [They encourage each other at work].
	SD3	Se llaman entre sí para tomar decisiones importantes. [They call each other to make important decisions].
	SD4	Evalúan conjuntamente el desempeño empresarial. [They jointly evaluate business performance].

**Table 4 behavsci-15-00130-t004:** Descriptive analysis of the items (n = 272).

Code	Average	Median	Standard Deviation	Asymmetry	Kurtosis
SE1	4.0809	4.0000	0.98741	−1.136	0.931
SE2	4.0993	4.0000	0.94561	−1.070	0.946
EL1	4.0368	4.0000	0.95591	−1.172	1.460
EL2	4.0331	4.0000	0.96565	−1.057	1.016
EL3	3.9301	4.0000	0.98639	−0.928	0.690
SD1	4.0588	4.0000	0.86987	−1.131	1.927
SD2	4.0846	4.0000	0.85273	−1.134	1.951
SD3	4.0846	4.0000	0.93132	−1.080	1.090
SD4	4.0221	4.0000	0.93688	−0.939	0.745

**Table 5 behavsci-15-00130-t005:** Exploratory Factor Analysis (EFA) pattern matrix: Own elaboration.

Item	Factor		
	1	2	3
SD3	0.980		
SD4	0.874		
SD2	0.807		
SD1	0.696		
EL1		0.908	
EL2		0.897	
EL3		0.794	
SE1			0.927
SE2			0.865

Extraction method: Maximum likelihood. Rotation method: Promax with Kaiser normalization.

**Table 6 behavsci-15-00130-t006:** Validation of the final measurement model with convergent reliability and validity.

Predictor	Items	Estimate	α	CR	AVE
Servant Leadership (SE)	SE1	0.914 ***	0.918	0.918	0.849
	SE2	0.929 ***			
Empowering Leadership (EL)	EL1	0.946 ***	0.956	0.957	0.880
	EL2	0.937 ***			
	EL3	0.932 ***			
Shared Leadership (SD)	SD1	0.911 ***	0.933	0.925	0.755
	SD2	0.914 ***			
	SD3	0.826 ***			
	SD4	0.821 ***			

Cronbach’s alpha (α) for all variables is > 0.7, the composite reliability (CR) > 0.60, and the mean-variance extracted (AVE) > 0.50; *** *p* < 0.001 (significance level), indicating a significant validity of the model.

**Table 7 behavsci-15-00130-t007:** Statistical goodness-of-fit indices of the scale to assess leadership style.

Measure	Threshold	Model 1		Model 2	
		Estimate	Interpretation	Estimate	Interpretation
CMIN	--	96.207	--	41.266	--
DF	--	24.000	--	23.000	--
CMIN/DF	Between 1 and 3	4.009	Acceptable	1.794	Excellent
CFI	>0.95	0.972	Excellent	0.993	Excellent
SRMR	<0.08	0.032	Excellent	0.023	Excellent
RMSEA	<0.06	0.105	Terrible	0.054	Excellent
PClose	>0.05	0.000	Not Estimated	0.369	Excellent

Note: CMIN = Chi square; DF = Degrees of freedom; SRMR = standardized root mean square residual; RMSEA = Root Mean Square Error of Approximation; CFI = comparative fit index. Model 2: e6–e7.

**Table 8 behavsci-15-00130-t008:** Discriminant validity.

Fornell–Larcker Criterion				Heterotrait–Monotrait Ratio (HTMT)	
	HE	HE	SD	Correlation	HTML
HE	**0.921**			SE-THE	0.755
HE	0.754	**0.938**		SE-SD	0.698
SD	0.714	0.752	**0.869**	EL-SD	0.748

## Data Availability

The original contributions presented in the study are included in the article, further inquiries can be directed to the corresponding author.

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
