# Peer review of "Leadership Styles in Non-Profit Institutions: An Empirical Study for the Validation and Reliability of a Scale in the Latin American Context"

_behavsci, 2025, doi:10.3390/bs15020130_

Round 1

Reviewer 1 Report

Comments and Suggestions for Authors

Dear Authors,

The topic of the paper is very interesting and relevant, especially the perspective on leadership styles in non-profit institutions, which highlights very significant aspects of the given subject.

However, I have some concerns regarding this paper.

First of all, Why didn't you use a standardized questionnaire and instead chose to create your own? Also, I'm not entirely sure why these particular constructs were chosen instead of others.

Second, the discussion is written rather briefly. It would be beneficial to cover a broader perspective and discuss each of the styles individually in more detail. Additionally, support the discussion with relevant literature sources.

Author Response

Reviewer 1

Author's Response

The topic of the article is very interesting and relevant, especially the perspective on leadership styles in non-profit institutions, which highlights very significant aspects of the topic in question.

Dear reviewer, we are deeply grateful for your comments, we have made efforts to make the manuscript worthy of this journal.

Why didn't they use a standardized questionnaire and instead decided to create their own? Also, I'm not entirely sure why these particular constructs were chosen instead of others.

Dear reviewer, thank you for your comment. It is perhaps worth mentioning that the purpose of this research was not focused on developing a new scale, but rather on validating an existing one (Cahyadi et al., 2022) in workers of non-profit entities for the context of Latin America. You can review the more detailed explanation in lines 198-212.

Secondly, the discussion is written quite briefly. It would be beneficial to cover a broader perspective and discuss each of the styles individually in more detail. Also, please support the discussion with relevant bibliographical sources.

Dear reviewer, thank you for your comment.  “5. Discussion” section has been improved, please review lines 292-322.

Reviewer 2 Report

Comments and Suggestions for Authors

This paper is written and referenced well and I have no suggestions for improvement.

I believe the papers is well written and because they have focused on learning styles they have used the usual researchers that you would expect to see from the literature. As leadership styles in education is not a current focus I can only state that how they have used the theorist in their case is applicable. I am a qualitative researcher, and their use of instrument design is not an area I am an expert on, however the arguments and discussion of findings are coherent and balance. I cannot comment on their choice of instruments and analysis. The paper is well referenced from the literature and the conclusions and note on limitations is accurate.

The main question is addressed using the literature and findings and have noted previously that the topic may be more popular in not for profits and in their geography. In school education we would argue strongly that these are now categorised as part of the skill set that a leader has or needs to improve in.

The conclusions are consistent with the evidence and arguments presented and the main question is addressed.

Author Response

Reviewer 2

Author's Response

This article is well written and referenced and I have no suggestions for improvement. I think the article is well written and as it has focused on learning styles it has used the usual researchers one would expect from the literature. As leadership styles in education are not a topic of current interest I can only state that the way they have used the theorist in their case is applicable. I am a qualitative researcher and their use of instrument design is not an area I am an expert in, however the arguments and discussion of findings are coherent and balanced. I cannot comment on their choice of instruments and analysis. The article is well referenced from the literature and the conclusions and note on limitations are accurate.

Dear reviewer, we are deeply grateful for your comments, we have made efforts to make the manuscript worthy of this journal.

The main question is addressed using the literature and findings and I have noted above that the topic may be more popular in non-profit organisations and their geography. In school education, we would strongly argue that they are now classified as part of the skill set that a leader has or needs to improve.

Dear reviewer, we are deeply grateful for your comments, we have made efforts to make the manuscript worthy of this journal.

The conclusions are consistent with the evidence and arguments presented and the main question is addressed.

Dear reviewer, we are deeply grateful for your comments, we have made efforts to make the manuscript worthy of this journal.

Reviewer 3 Report

Comments and Suggestions for Authors

This is supposed to eb a scale development paper but unfortunately it fell flat.

1. The authors did not clearly justify why this new scale is needed other than saying that no research has been done in teh context.

2. Scale development papers usually cites the paper by Churchill for its development process unfortunately this paper did not do that.

3. How were the items derived?

4. I doubt any leadership style can be assessed using 2, 3 and 4 items, this is unheard of in the literature.

5. Correlating erro terms are also not acceptable.

6. The paper looks more like an exercise in statistics rather than a scale development paper.

7. Scale development papers should also assess nomological validity.

Author Response

Reviewer 3

Author's Response

The authors did not clearly justify why this new scale is needed, beyond saying that no research has been done in that context.

Dear Reviewer, Thank you for your comments. Your observation has been taken into account. Please review lines 68-93 & 145-160.

Scale development documents usually cite the Churchill paper for their development process, but unfortunately this document did not.

Dear reviewer, thank you for your comment. It is perhaps worth mentioning that the purpose of this research was not focused on developing a new scale, but rather on validating an existing one (Cahyadi et al., 2022) in workers of non-profit entities for the context of Latin America. You can review the more detailed explanation in lines 198-212.

How did they get the elements?

Dear reviewer, thank you for your comment. Section “3.3. Measurement scale and back translation” has been improved. You can find a more detailed explanation on lines 201-209.

I doubt that any leadership style can be assessed using 2, 3 and 4 items, this is unheard of in the literature.

Dear reviewer, thank you for your comment. “5. Discussion”, has been improved. You can review the more detailed explanation in lines 301-322.

The document reads more like a statistical exercise than a scale development document.

Dear reviewer, thank you for your comment. It may be worth mentioning that the purpose of this research was not focused on developing a new scale, but rather validating an existing one (Cahyadi et al., 2022) in workers of non-profit entities for the context of Latin America. However, your comment has been taken into account and you can review the more detailed explanation in lines 198-212. In addition, the sections: “2.2. Scales to evaluate leadership styles” and “5. Discussion” (shaded in yellow) have been strengthened, to improve the document on this point.

Scale development documents should also assess nomological validity.

Dear reviewer, we deeply appreciate your comment on the need to assess nomological validity in scale development. We recognize that this is a critical step to ensure that the scale dimensions are aligned with a sound theoretical framework and that their relationships with other expected constructs are consistent. In this study, by focusing on the validation of the scale to measure leadership styles, we have addressed construct validity (convergent and discriminant validity) and internal reliability. However, we accept that nomological validity was not assessed since it has not been collected with other constructs, in that sense, we comment on this point in the limitations and that it is recommended to do this in future research.

Round 2

Reviewer 3 Report

Comments and Suggestions for Authors

The revisions are acceptable .

Author Response

Ref.: Manuscript ID: behavsci-3266778

Leadership Styles in non-profit Institutions: An Empirical Study for the Validation and Reliability of a Scale in the Latin American Context

Dear Reviewer,

Thank you very much for your comments reported above, they helped us greatly to improve the manuscript. We appreciate the time you spent on this process.

Thank you very much for your time.

Sincerely,
